# Active Classification with Few Queries under Misspecification

**Vasilis Kontonis**[*]
UT-Austin
vasilis@cs.utexas.edu

**Mingchen Ma**[*]
University of Wisconsin-Madison
mingchen@cs.wisc.edu

**Christos Tzamos**[*]
University of Athens and Archimedes AI
ctzamos@gmail.com

## Abstract

We study pool-based active learning, where a learner has a large pool $S$ of unlabeled examples and can adaptively ask a labeler questions to learn these labels. The goal of the learner is to output a labeling for $S$ that can compete with the best hypothesis from a given hypothesis class $\mathcal{H}$. We focus on halfspace learning, one of the most important problems in active learning.

It is well known that in the standard active learning model, learning the labels of an arbitrary pool of examples labeled by some halfspace up to error $\epsilon$ requires at least $\Omega(1/\epsilon)$ queries. To overcome this difficulty, previous work designs simple but powerful query languages to achieve $O(\log(1/\epsilon))$ query complexity, but only focuses on the realizable setting where data are perfectly labeled by some halfspace. However, when labels are noisy, such queries are too fragile and lead to high query complexity even under the simple random classification noise model.

In this work, we propose a new query language called threshold statistical queries and study their power for learning under various noise models. Our main algorithmic result is the first query-efficient algorithm for learning halfspaces under the popular Massart noise model. With an arbitrary dataset corrupted with Massart noise at noise rate $\eta$, our algorithm uses only $\operatorname{poly}\log(1/\epsilon)$ threshold statistical queries and computes an $(\eta + \epsilon)$-accurate labeling in polynomial time. For the harder case of agnostic noise, we show that it is impossible to beat $O(1/\epsilon)$ query complexity even for the much simpler problem of learning singletons (and thus for learning halfspaces) using a reduction from agnostic distributed learning.

## 1 Introduction

Obtaining labeled examples is often challenging in applications as querying either human annotators or powerful pre-trained models is time-consuming and/or expensive. Active learning aims to minimize the number of labeled examples required for a task by allowing the learner to adaptively select for which examples they want to obtain labels. More precisely, in pool-based active learning, the learner has to infer the labels of a pool $S$ of $n$ unlabeled examples, and can adaptively select an example $x \in S$ and ask for its label.

Even though it is known that active learning can exponentially reduce the number of required labels, this is unfortunately only true in very idealized settings such as datasets labeled by one-dimensional

---

[*]Equal Contribution

38th Conference on Neural Information Processing Systems (NeurIPS 2024).

thresholds or structured high-dimensional instances (e.g., Gaussian marginals) [17, 4, 6, 7, 3, 19, 23]. It is well-known that without such distributional assumptions, even for linear classification in 2 dimensions, active learning yields no improvement over passive learning [15, 16].

**Active Learning with Queries**  To bypass the hardness results and establish learning without restrictive distributional assumptions [5, 33, 30, 31, 47, 10] introduce enriched queries, where the learner is allowed to make more complicated queries. Broadly speaking, there are two lines of work that study active learning with enriched queries. The first one designs queries based on the structure of the hypothesis class it aims to learn. For example, [33] design comparison queries for learning halfspaces in 2 dimensions, [31] design same-leaf queries for learning decision trees, and [8] design derivative queries to learn polynomial threshold functions. The success of these queries heavily depends on the relation between the hypothesis class and the properties of the queries and thus a completely new query language has to be designed if the learning problem gets changed. The other line of work such as [5, 10, 44] study mistake-based queries, asking questions like if a positive example exists in a given set. These works break down a complicated learning problem into a small number of simple statistical tasks that require only very few rounds of interactions with a labeler who knows the hidden labels and can easily solve these tasks. These learning models can be formally summarized as follows.

**Definition 1.1** (Active Learning with Enriched Queries)**.** *Given a (multi)set of $n$ unlabeled examples $S \subseteq \mathcal{X}$ over a domain $\mathcal{X}$, a learner $\mathcal{A}$ wants to output a hypothesis $\hat{f} : \mathcal{X} \to \{\pm 1\}$ by adaptively submitting binary queries to a labeler who knows the hidden labels of the examples $S$. Each query $q : 2^{S \times \{\pm 1\}} \to \{0, 1\}$ is a function that takes a subset of examples in $S$ together with their unknown labels as input and outputs a number in $\{0, 1\}$. If $f(x) : S \to \{\pm 1\}$ is the unknown labeling function, the learner aims to make the error of $\hat{f}$,*

$$\mathrm{err}(f) := \frac{1}{n} \sum_{x \in S} \mathbf{1}(\hat{f}(x) \neq f(x))$$

*as small as possible compared to a target class of binary hypotheses $\mathcal{H}$ over $\mathcal{X}$.*

In the *realizable setting* where the unknown labeling function belongs to the target hypothesis class $\mathcal{H}$, several non-trivial hypothesis classes including the class of halfspaces have been proven efficiently learnable using only a logarithmic number of rounds of interactions. For example, [10] shows for any set $S$ of $n$ examples satisfying $\gamma$-margin condition with respect to the underlying halfspace $h^*$, one can use $O(d \log(d/\gamma))$ seed queries, which returns an example with a specified label in a given region, to perfectly learn labels of all examples in $S$ efficiently. More recently, [44] shows that for an arbitrary set of $n$ examples labeled by an arbitrary halfspace, efficiently learning all the labels only requires $\tilde{O}(d^3 \log(n))$ region queries, which ask if an example with a specified label exists in a given region.

While these results show that mistake-based queries are extremely powerful in the realizable case, they fail to capture most practical cases where there is typically model misspecification or errors in the data. In fact, it is not hard to see that even under tiny amounts of noise, these mistake-based queries become essentially unusable and can be simulated by label queries. For example, if the labels of the examples are flipped with probability $\eta$, say 10% (random classification noise [2]), then a region query over a region that contains more than 10 examples will return "yes" with extremely high probability, which provides no information to the learner. In fact, even for the more powerful seed query used in [10], where in addition an example with the specified label is returned, [5] shows that if an algorithm can learn the labels of a set of examples $S$ with error $\eta + \epsilon$, with ground truth in a given hypothesis class $H$ corrupted by $\eta$-level random classification noise using $M(\epsilon)$ queries, then one can simulate such an algorithm using $M(\epsilon)/\eta$ label queries. Such a result implies even for simple hypothesis classes such as the class of intervals in real line or the class of halfspaces in 2 dimensions, one needs at least $\Omega(1/\epsilon)$ seed/region queries to learn the labels of examples in a given dataset to error $\eta + \epsilon$. The gap between the realizable setting and the noise setting motivates the following natural questions:

*Can we design a simple noise-tolerant query language, that allows learning non-trivial hypothesis classes efficiently with few queries?*

Similar to [10, 35], in this work, we focus on the class of halfspaces, one of the most important hypothesis classes in active learning.

## 1.1 Learning Model And Our Contribution

We propose a new query language called Threshold Statistical Queries (TSQ), which generalizes the region queries used in [10, 35] and study its power for learning halfspaces under noise.

**Definition 1.2** (Threshold Statistical Queries (TSQ)). *Let $S$ be a set of examples in a domain $\mathcal{X}$ with a corresponding labeling function $f : S \to \{\pm 1\}$. A threshold SQ query $q(\phi, \tau)$ takes as input a function $\phi(x, y)$ over $S \times \{\pm 1\}$ and a threshold $\tau \in \mathbb{R}$, and answers whether $\sum_{x \in S} \phi(x, f(x)) \geq \tau$.*

TSQ is a simple generalization of region queries and vanilla label queries. For a region $U \subseteq S$ and a target label $a \in \{\pm 1\}$, if $\phi(x, y) = \mathbf{1}(x \in U \wedge f(x) = a)$, then $q(\phi, 1)$ is exactly the region query, where it checks if at least one example in $U$ has label $a \in \{\pm 1\}$. Furthermore, if $|U| = 1$, then $q(\phi, 1)$ is exactly the classic label queries.

Our goal is to study the power of TSQ for active learning under different label noise models. We consider 3 progressively more challenging noise models commonly studied in the literature: Random, Massart and Adversarial.

**Definition 1.3** (Active Learning under Label Noise). *Let $\mathcal{H}$ be a hypothesis class over domain $\mathcal{X}$. Let $S \subseteq \mathcal{X}$ be a (multi)set of $n$ examples and $h^* \in \mathcal{H}$ be a ground truth function. For a parameter $\eta \in [0, 1/2)$, the labeling function $f(x)$ over $S$ is generated in the following way under the three label noise models.*

- ***Random Classification Noise (RCN) [2]:** For each $x \in S$, $f(x)$ is $-h^*(x)$ with probability $\eta$ and $h^*(x)$ otherwise.*

- ***Massart Noise [41]:** For each $x \in S$, $f(x) = -h^*(x)$ with some unknown probability $\eta(x) \leq \eta$ and $h^*(x)$ otherwise.*

- ***Adversarial Label Noise:** For an unknown subset $S'$ containing $\eta$ fraction of examples from $S$, $f(x) = -h^*(x)$ for all $x \in S'$ and $f(x) = h^*(x)$ for all $x \in S \setminus S'$.*

*Given the unlabeled examples $S$, and an error parameter $\epsilon \in (0, 1)$, the goal of the learner is to output a labeling $\hat{f}$ over $S$ such that with high probability $\mathrm{err}(\hat{f}) \leq \eta + \epsilon$*

We remark that in our model, after the labeling $f(x)$ is generated, the label of each example $x \in S$ will be fixed throughout the learning process, also known as persistent noise. This means if an algorithm keeps querying the label of the same example, it will receive the same answer. Furthermore, under the Random classification noise/Massart noise model, we will assume the size $n$ of the dataset $S$ is large enough ($\mathrm{poly}(d, 1/\epsilon)$), because if $n$ is small we have even no guarantee on the error of the ground truth hypothesis. Our main algorithmic result is the first distribution-free halfspace learning algorithm that achieves both computational efficiency and query efficiency under the (persistent) Massart noise model and the Random Classification Noise model.

**Theorem 1.4.** *Let $\mathcal{H} = \{h(x) = \mathrm{sign}(w \cdot x) \mid w \in S^{d-1}\}$ be the class of halfspaces over $\mathbb{R}^d$. Given parameters $\epsilon, \delta \in (0, 1)$, a set $S$ of $n = \mathrm{poly}(d, 1/\epsilon, \log(1/\delta))$ examples in $\mathcal{X}$ and TSQ query access to an unknown labeling corresponding to a ground truth hypothesis $h^* \in \mathcal{H}$ corrupted with Massart noise $\eta \in [0, 1/2)$, we can compute in $\mathrm{poly}(n)$ time a labeling $\hat{f}$ such that $\mathrm{err}(\hat{f}) \leq \eta + \epsilon$, with probability at least $1 - \delta$, making $\tilde{O}(d^3 \log^3(1/\epsilon))$ threshold SQ queries.*

Importantly, unlike [10], we make no structure assumption over the dataset $S$, and the query complexity of our algorithm qualitatively matches the query complexity obtained by [35], which only holds in the realizable setting. Theorem 1.4 shows a sharp separation between standard active learning/region queries which require $\mathrm{poly}(1/\epsilon)$ query complexity and threshold statistical queries where $\mathrm{poly}\log(1/\epsilon)$ query complexity suffices under the Massart noise and Random classification noise models. Furthermore, we will discuss in Appendix A that the TSQs we use in the algorithm have very simple strictures. A natural question is whether TSQ can tolerate more complex noise.

Our second main result is a negative result showing that even using TSQ, it is still hard to achieve query efficiency under the adversarial label noise even for simpler hypothesis classes such as the class of singleton and the class of intervals and thus for the class of halfspaces. Formally, we have the following theorem.

**Theorem 1.5.** *Let $\mathcal{H}$ be the class of singleton functions over the domain $\mathcal{X} = \mathbb{N}$. For every $\epsilon \in (0, 1)$ and $m > \Omega(1/\epsilon)$, there is a set $S$ of $m$ examples over $\mathcal{X}$ and a labeling function $f$ for $S$ such that*

*any learning algorithm $\mathcal{A}$ that makes less than $\tilde{O}(1/\epsilon)$ TSQs must output, with probability at least $1/3$, a labeling function $\hat{f}$ with error $\mathrm{err}(\hat{f}) > \mathrm{opt} + \epsilon$, where $\mathrm{opt} = \min_{h \in \mathcal{H}} \mathrm{err}(h)$.*

As we can always embed an instance of learning singleton into an instance of learning a 2-dimensional halfspace, Theorem 1.5 also implies a $\tilde{\Omega}(1/\epsilon)$ query complexity for agnostic learning halfspaces with TSQ. This shows a sharp separation of the performance of TSQ under different noise models and leaves designing more robust query languages as an important future direction. From a technical perspective, unlike usual approaches in the active learning literature which explicitly construct hard instances [16, 28], we obtain our result via reduction from the agnostic distributed learning problem studied by [32] for which a communication complexity lower-bound has been established. To the best of our knowledge, this is the first result that connects distributed learning and active learning, two seemingly unrelated learning models.

Though, Theorem 1.5 shows that agnostic learning up to error $\mathrm{opt} + \epsilon$ cannot be achieved in a query efficient way, inspired by the work of [5], it is possible to use only $\tilde{O}(d \log(1/\epsilon))$ TSQ to learn the label of a dataset up to error $O(\mathrm{opt}) + \epsilon$ for every hypothesis class with finite VC dimension, though the running time of the algorithm is exponential. Such a result might be of independent interest as how to efficiently learn a hypothesis up to error $O(\mathrm{opt}) + \epsilon$ have already been studied in many agnostic learning literature such as [13, 14, 22]. We leave the proof of Theorem 1.6 to Appendix C due to the space limit.

**Theorem 1.6.** *Let $\mathcal{X}$ be the space of examples and $\mathcal{H}$ be a hypothesis class over $\mathcal{X}$ with VC-dimension $d$, there is an algorithm such that for every $\epsilon, \delta \in (0, 1)$, for every set $S$ of $n$ examples, and for every labeling function $f(x)$, it makes $\tilde{O}(d \log(1/\epsilon))$ TSQs and outputs a labeling $\hat{f}$ such that with probability $1 - \delta$, $\mathrm{err}(\hat{f}) \leq O(\mathrm{opt}) + \epsilon$, where $\mathrm{opt} = \min_{h \in \mathcal{H}} \mathrm{err}(h)$.*

## 1.2 Related Works

**Active Learning with Mistake-Based Queries**   Learning with mistake-based queries has a long history [1, 39, 5, 10]. A typical mistake-based query can be understood as follows. A learner selects a subset of examples $T \subset \mathcal{X}$ and proposes a possible labeling for them to a labeler. The labeler will return an example $x \in T$ labeled incorrectly by the learner or return nothing when every example in $T$ is labeled correctly. Beyond being quite successful in theory, mistake-based queries also have wide applications in commercial systems [11, 25]. In the realizable setting, it has been well-known that such queries can be used to implement the Halving algorithm [38] and achieve $O(d \log(1/\epsilon))$ query complexity for hypothesis classes of VC dimension $d$. However, it is only until very recently [10, 35] that people know how to use these queries to design algorithms that achieve both computational efficiency and query efficiency. In the noisy setting, [5] shows that even under random classification noise, it is impossible to use such queries to do query-efficient learning even for very simple classes. In this work, we propose TSQ as a robust generalization of these queries.

**Statistical Query Learning Model**   Close to our threshold statistical learning model (TSQ) is the classic statistical learning model (SQ) proposed by [34]. SQ was originally designed to overcome random classification noise but has numerous applications in learning theory literature as a refinement of the PAC learning model which captures most algorithms used in practice. It has been used as a tool for obtaining efficient learning algorithms robust to noise [9] and as an evidence of computational difficulty of a statistical problems [21]. In the SQ model, the learner has no direct access to any example but can evaluate the expectation $\mathbf{E}_{(x,y) \sim D} \phi(x, y)$ for an arbitrary bounded function $\phi(x, y)$ within accuracy $\delta$. This means in SQ model, a learning algorithm should consider both the time used for computing $\phi(x, y)$ but also have to consider the final accuracy. On the other hand, a TSQ is a boolean function of the unlabeled examples and their hidden labels. No matter the complexity, any TSQ, $q(\phi)$ can be computed by the labeler accurately in time at most $O(n)$. Furthermore, as in SQ model, a learner has no access to individual examples, SQ learning does not naturally fit in the active learning model. One even cannot implement classic active learning algorithms such as CAL or Halving [27, 28] in the SQ model. As opposed to SQ, our TSQ model is more powerful as it can isolate individual examples and thus fills such a gap. We remark that this more powerful type of access is not needed for Theorem 1.4 and can be implemented with SQ queries of $\mathrm{poly}(\epsilon)$ accuracy.

**Learning Halfspaces with Massart Noise**   Active learning for halfspaces under Massart noise also has a long history. Many works [4, 46, 3, 48] design learning algorithms that achieve both

computational efficiency and query efficiency under structured distributions such as the uniform distribution over the unit sphere, the Gaussian distribution, and log-concave distributions. On the other hand, without distributional assumptions, learning under Massart noise is much more challenging. Computationally efficient learning algorithm for learning halfspaces under Massart noise [18, 12, 20] were only recently discovered for passive learning. Our algorithm is the first one that works in an active learning setting and achieves both computational efficiency and query efficiency.

## 2 Learning Halfspaces under Massart Noise

In this section, we present Theorem 1.4, our main algorithmic result. The full proof is left at Appendix A. To start with, we give a high-level overview of how our algorithm works. Similar to previous works on distribution-free learning halfspaces [9, 18, 35], our learning algorithms recursively run two subroutines over the dataset $S$. The first subroutine is a weak learning algorithm that works under structured datasets $S'$. More specifically, we assume that all points in the dataset have unit norm and for every direction $w \in S^{d-1}$, there are at least $\Omega(1/d)$ fraction of the examples $x$ in $S'$ such that $|w \cdot x| \geq \Omega(1/\sqrt{d})$. Intuitively, the regions $\{x \in S' \mid |w \cdot x| \geq \Omega(1/\sqrt{d})\}$ correspond to examples for which a halfspace with normal vector $w$ is more confident about the label. If $S'$ contains a non-trivial fraction of examples in $S$ and we can run a weak learning algorithm over $S'$ to learn a vector $w$ that has a classification error $\eta + \epsilon$ over $\{x \in S' \mid |w \cdot x| \geq \Omega(1/\sqrt{d})\}$, we are able to label a non-trivial fraction of examples in $S$ with a low error. In Section 2.1, we will design such a learning algorithm that is robust to Massart noise and achieves query efficiency and computational efficiency simultaneously. However, in general, it is not always possible to find a large enough subset from $S$ that is in an approximate radially isotropic position. Forster's transform [26], a powerful preprocessing technique can be used to solve this issue. Given any set of $n$ examples in $\mathbb{R}^d$, we can always use Forster's transform to find a subset of $kn/d$ examples that lie in a $k$ dimensional subspace such that after a non-linear transformation, the transformed examples are in an approximate radially isotropic position. This implies that if we can implement our weak learning algorithm over the transformed data, each round, we are able to label $1/d$ fraction of the whole dataset with small error and thus after $d \log(1/\epsilon)$ rounds of weak learning, only $\epsilon$ fraction of the examples are unlabeled. In Section 2.2, we will show how to use Forster's transform to select a large fraction of the dataset for the weak learning algorithm and how to implement the weak learning algorithm over the transformed dataset using TSQ. Furthermore, we want to point out that the TSQs we use in our algorithms have very simple structures. We leave the discussion in detail in Appendix A.

### 2.1 A Weak Learning Oracle

In this section, we present our weak learning algorithm, Algorithm 1, which plays a central role in Theorem 1.4. Our main algorithmic result in this section is the following theorem, the proof of which can be found in Appendix A.

**Theorem 2.1.** *Let $V \subseteq \mathbb{R}^d$ be a subspace of dimension $k$ and $S \subset V$ be a set of $n = \mathrm{poly}(k, 1/\epsilon, \log(1/\delta))$ examples with unit length. Let $h^*(x) = \mathrm{sign}(w^* \cdot x), w^* \in B_1^k$ be the ground truth hypothesis. If for every unit vector $w \in B_1^k$, at least $1/4d$ fraction of examples $x \in S$ satisfy $|w \cdot x| \geq 1/(2\sqrt{k})$, and $u \cdot w^* \geq 1/(4\sqrt{k})$, then under the Massart noise model, for every ground truth , with probability at least $1 - \delta$, Algorithm 1 outputs $(S', \hat{f}_{S'})$ such that $|S'| \geq n/(4k)$ and $\hat{f}_{S'}$ has error at most $\eta + \epsilon$ over $S'$, using $\tilde{O}(d^2 \log^2(1/\epsilon))$ TSQ, in $\mathrm{poly}(n, k)$ time.*

To understand why Algorithm 1 is robust to Massart noise, we need to understand why such a problem is difficult. Let $S$ be a subset of $n$ example in an approximate radially isotropic position. Take the algorithm in [35] as an example. Such an algorithm uses a modified perception algorithm to learn some $w$ that can perfectly classify all examples that have a large margin with respect to it. Namely, in each round, either the current hypothesis $w_i$ perfectly classifies a large fraction of examples or seed/region queries are used to quickly find an example in that region that is misclassified by $w_i$, which will be fed to the perception algorithm and improve $w_i$. In the noisy setting, however, every example $x$ has a constant probability of being misclassified by $w_i$. This implies we need to use queries to find a "point" where $w^*$ and $w_i$ disagree. To do this, we associate each example $x$ in the region $S_i = \{x \in S \mid |w \cdot x| \geq \Omega(1/\sqrt{d})\}$, with a variable $Y_x \in \{0, 1\}$, where $Y_x = 1$ if $\mathrm{sign}(w_i \cdot x) \neq y(x)$ and 0 otherwise. If the noise $\eta(x) = \eta$ for every example $x$ i.e. Massart noise

---

**Algorithm 1** WEAKLY LEARNING HALFSPACES (Labeling $1/d$ fraction of examples via TSQ)

---

**Input:** $\epsilon, \delta \in (0,1)$, subspace $V \subseteq \mathbb{R}^d$ of dimension $k$, $S \subset V$ of $n$ examples with unit length, $u$, a unit vector in $V$

**Output:** $S' \subseteq S$ a subset of examples, $\hat{f}_{S'} : S' \to \{\pm 1\}$ a labeling for $S'$

Let $P_0 = \{x \in B_1^k \mid u \cdot x \geq 1/(4\sqrt{k})\}$, where $B_1^k$ is the unit ball in $V$.

Compute $x_0 \in P_0$ using Vaidya's algorithm by Theorem 2.3.

**for** $i = 0, \ldots, \tilde{O}(k)$ **do**

    Let $w_i = x_i / \|x_i\|$

    Check if over $S_{w_i} = \{x \in S \mid |w_i \cdot x| \geq \frac{1}{2\sqrt{k}}\}$, $w_i$ has error larger than $\eta + \epsilon$ via TSQ

    If $w_i$ has error less than $\eta + \epsilon$ over $S_{w_i}$, **return** $(S_{w_i}, \text{sign}(w_i \cdot x))$ and stop the algorithm

    Draw a random set $U$ from $S_{w_i}$ of size $m = \tilde{O}(k^2/\epsilon^2)$.

    For each $x \in S_{w_i}$, define

$$\phi(x,y) = (Y_x(y) - \eta)/(w_i \cdot x),$$

where $Y_x = 1$ if $y \neq \text{sign}(w_i \cdot x)$ and $Y_x = 0$ otherwise.

    Use $\tilde{O}(d)$ TSQ to do binary searches along each coordinate and find some $c_i$ such that $\left\| c_i - \frac{1}{m} \sum_{x \in U} \phi(x,y)x \right\|_\infty \leq \epsilon/(8k^2)$.

    Feed Vaidya's algorithm by $((c_i - \epsilon u/4)^t, 0)$ and compute $(P_{i+1}, x_{i+1})$.

Report Fail if nothing has been returned

---

model degenerates to the random classification noise model, then consider the following point

$$\bar{x} = \sum_{x \in S_i} (Y_x - \eta)x = \sum_{x \in S_+} (Y_x - \eta)x + \sum_{x \in S_-} (Y_x - \eta)x,$$

where $S_+$ is the subset of examples in $S_i$ where $w_i$ agrees with $w^*$ and $S_- = S_i \setminus S_+$. For each $x \in S_+$, $\mathbf{E} Y_x = \eta$, while for every $x \in S_-$, $\mathbf{E} Y_x = 1 - \eta$. This implies that in expectation, $\mathbf{E} \bar{x} = (1 - 2\eta) \sum_{x \in S_-} x$. After properly scaling, this gives a point in $S_i$ where $w_i$ and $w^*$ disagree due to the convexity of the problem and thus serves as a counter-example to run the perception algorithm. In particular, since the contribution of each example $x$ only depends on its true label, we can draw random samples from $S_i$ and use TSQ along each coordinate to approximately find $\bar{x}$ up to high accuracy using very few queries via binary search.

However, for Massart noise, this is not the correct way to design a learning algorithm. This is because $\eta(x)$ is non-uniform over each $x$. For simplicity, we assume $w_i \cdot x > 0$ for each $x \in S_i$. As $\eta(x) \leq \eta$, a simple calculation shows that $E\bar{x} \cdot w^* \leq 0$, where the randomness only comes from the Massart noise. The hope is that if $w_i$ has an error $\eta + \epsilon$ over $S_i$, then $\bar{x} \cdot w_i$ is larger than some positive number so that we find a counter-example. This is unfortunately not true. Because $w_i \cdot x$ are different and $\eta(x)$ are different, even if the error is large, some of the examples with large margins could force $\bar{x}$ points to the opposite direction, making $\bar{x} \cdot w_i \leq 0$ as well. To overcome this issue, we consider using a slightly more complicated statistic here, where we define

$$\bar{x} := \sum_{x \in S_i} (Y_x - \eta) \frac{x}{w_i \cdot x}$$

instead. Such a point is still easy to approximate up to error $\epsilon$ with only $d \log(1/\epsilon)$ TSQs, because it is each to compute $w_i \cdot x$ for each $x \in S_i$. But more importantly, when $w_i$ has an error larger than $\eta + \epsilon$, in expectation $w_i$ and $w^*$ will always disagree on $\bar{x}$ because

$$\frac{1}{|S_i|} w_i \cdot \bar{x} = \frac{1}{|S_i|} \sum_{x \in S_i} (Y_x - \eta) \frac{x}{w_i \cdot x} \cdot w_i = \frac{1}{|S_i|} \sum_{x \in S_i} (Y_x - \eta) > \epsilon. \tag{1}$$

Furthermore, as $w_i \cdot x$ is large for every $x \in S_i$, $\bar{x}$ has a bounded norm and thus can serve as a counter-example. A technical issue here is that the inequality $\mathbf{E} \bar{x} \cdot w^* \leq 0$ is quite fragile, due to the randomness of the Massart noise, it is impossible to guarantee $\bar{x} \cdot w^* \leq 0$ actually holds after the labeling being fixed. This issue can be fixed using the following trick. Before run the learning algorithm, we will randomly sample a unit vector $u$. We know from [45] that with constant probability

$u \cdot w^* > 1/\sqrt{d}$ and thus by shifting $\bar{x}$ a little towards $-u$, this will give us a counter example and guarantee the whole algorithm succeeds with a constant probability.

Though, we find a counter-example and can use it to run a perception algorithm in a similar way to [35], this cannot give us a good query complexity. This is because (1) can only guarantee $\bar{x} \cdot w_i > \epsilon$, which requires to run the perception algorithm for $O(1/\epsilon^2)$ rounds to converge to a good hypothesis. Thus, we will solve this problem using Vaidya's cutting plane method. We want to remind the reader of the following convex feasibility problem, which is closely related to our halfspace learning problem.

**Definition 2.2** (Convex Feasibility Problem). *Let $K \subset \mathbb{R}^d$ be a convex body. A separation oracle with respect to $K$ is a function on $\mathbb{R}^d$ such that for any input $x \in \mathbb{R}^d$, if $x \in K$, then it reports "yes", otherwise it outputs some $(c^t, b) \in \mathbb{R}^{d+1}$ such that for every $y \in K$, $c \cdot y \geq b$ but $c \cdot x \leq b$. Assuming $K \subseteq B_1^d$, given a separation oracle with respect to $K$ and $\epsilon \in (0,1)$, the convex feasibility problem asks to either find some $x \in K$ or prove that $K$ does not contain a ball of radius $\epsilon$.*

There exists a long line of research for solving the convex feasibility problem for example, [43, 40, 37]. We will use these algorithms as a subroutine of our learning algorithm.

**Theorem 2.3** (Vaidya's Algorithm). *Let $K \subset P_0 \subseteq B_1^d$ be an unknown convex body. Vaidya's algorithm solves the convex feasibility problem for $K$ as follows. In round $i$, it maintains a convex body $K \subseteq P_i \subseteq P_0$ and a point $x_i \in P_i$ and sends $x_i$ to the separation oracle of $K$. If the oracle returns "yes", then it claims $x_i \in K$, otherwise it computes in $\mathrm{poly}(d, \log(1/\epsilon))$ time a pair of $(P_{i+1}, x_{i+1})$ based on $(c_i^t, b_i)$ the return of the separation oracle. In particular, after $T = \tilde{O}(d \log(1/\epsilon))$ rounds, $P_T$ does not contain a ball of radius $\epsilon$.*

Let the unknown convex body $K$ that we want to solve for the convex feasibility problem be a ball of radius $\epsilon$ around $w^*$ and we want to run Vadidya's algorithm to find some $w_i$ close to $w_*$. Consider the $P_i$ maintained by Vadiya's algorithm. As with constant probability $w^* \cdot u \geq 1/\sqrt{d}$ as we mentioned earlier, we can guarantee that $0 \notin P_i$. Let $x_i$ be the point used by Vadidya's algorithm. Then we will check the error of $w_i = x_i / \|x_i\|$ over $S_i$ is large, which can be done with a single TSQ. If the error is less than $\eta + \epsilon$, we are done. Otherwise, we use $\tilde{O}(d \log(1/\epsilon))$ TSQ to approximately find a counter example $\bar{x}$ for $w_i$. Importantly, the halfspace $\bar{x} \cdot w \geq 0$ separate $x_i$ and any $w \in K$. This will make it possible to run the next round of Vadiya's algorithm. Since we only care about examples that have margin $\Omega(1/\sqrt{d})$ with respect to $w_i$, when $w_i$ is within a ball of radius $1/\mathrm{poly}(d)$ centered at $w^*$, every example in $S_i$ is agreed by $w_i$ and $w^*$ and thus $w_i$ is guaranteed to have error at most $\eta + \epsilon$. Furthermore, in each round, Vadiya's algorithm shrinks the volume of $P_i$ by a constant factor, and after at most $\tilde{O}(d \log(1/\epsilon))$ rounds, we are guaranteed to find a good hypothesis. This gives a weak learning algorithm with a desired query complexity.

## 2.2 From Weak Learning to Strong Learning

We leave the formal analysis of the algorithm to Appendix A and discuss two technical issues raised in designing Algorithm 2. First, as required in Theorem 2.1, the dataset $S$ should be large enough and satisfy the structured assumption. Thus, to run Algorithm 1, we need to recursively select a dataset of enough size that satisfies the structured assumption from the data we have not labeled. In fact, the structured assumption can be fulfilled by a dataset that is in approximate radially isotropic position.

**Definition 2.4** (Approximate Radially Isotropic Position). *Let $S$ be a multiset of non-zero points in $\mathbb{R}^d$, we say $S$ is in $\epsilon$-approximate radially isotropic position, if for every $x \in S$, $\|x\| = 1$ and for every $u \in S^{d-1}$, $\sum_{x \in S} (u \cdot x)^2 / |S| \geq 1/d - \epsilon$.*

**Lemma 2.5.** *Let $S$ be a multiset of non-zero points in $\mathbb{R}^d$ that is in $1/2d$-approximate radially isotropic position. Then for every $u \in S^{d-1}$, we have $\mathbf{Pr}_{x \sim S}\left(|u \cdot x| \geq 1/2\sqrt{d}\right) \geq 1/4d$.*

Recent results show that for any dataset $S$, one can efficiently find a non-trivial fraction of the data and a non-linear transformation such that after the transform, the data are in approximate radially isotropic position.

**Theorem 2.6** (Approximate Forster's Transform [24]). *There is an algorithm such that given any set of $n$ points $S \subseteq \mathbb{R}^d \setminus \{0\}$ and $\epsilon > 0$, it runs in time $\mathrm{poly}(d, n, \log 1/\epsilon)$ and returns a subspace $V$ of $\mathbb{R}^d$ containing at least $\dim(V)/d$ fraction of points in $S$ and an invertible matrix $A \in \mathbb{R}^{d \times d}$ such*

---

**Algorithm 2** STRONG LEARNING HALFSPACES (Label $S$ with few queries up to $\eta + \epsilon$ error )

---

**Input:** $\epsilon, \delta \in (0, 1)$, $S \subset \mathbb{R}^d$ of $n$ examples
**Output:** $\hat{f} : S \to \{\pm 1\}$ a labeling for $S$
$L \leftarrow \emptyset$, $n \leftarrow |S|$
**while** $|S| > \epsilon n/2$ **do**
    Apply Theorem 2.6 to $S$ with $\epsilon = 1/2d$ to obtain a matrix $A$ and a $k$-dimensional subspace $V$
    Use a single TSQ to check if constant hypothesis $+1(-1)$ has error $\eta + \epsilon$ over $S \cap V$
    **if** constant hypothesis has error at most $\eta + \epsilon/2$ over $S \cap V$ **then**
        Define $\hat{f}$ to be the constant over $S' = S \cap V$
        $S \leftarrow S \setminus S'$
    **else**
        Run Algorithm 1 over input parameter $\epsilon/2, \delta/\mathrm{poly}(d, \log(1/\epsilon))$, $V$, $f_A(S \cap V)$ and a
random unit vector $u \in V$ until some $(S', \hat{f}_{S'})$ is output.
          $\triangleright$ Though Algorithm 1 is run over the transformed dataset $f_A(S \cap V)$, each TSQ can be
simulated over the original data as $F_A(x)$ preserves the ground truth label.
        Define $\hat{f}(x) = \hat{f}_{S'}(F_A(x)), \forall x, F_A(x) \in S'$
        $S \leftarrow S \setminus S'$
Define $\hat{f} = 1$ for the rest of $\epsilon n/2$ examples in $S$
**return** $\hat{f}$

---

that $F_A(S \cap V)$ is in $\epsilon$-approximate radially isotropic position up to isomorphic to $\mathbb{R}^{\dim(V)}$, where $F_A(S \cap V) = \{F_A(x) := Ax/\|Ax\| \mid x \in S \cap V\}$.

Combine Theorem 2.6 and Lemma 2.5, we know that given any set of $n$ examples $S \subseteq \mathbb{R}^d$, we can find a subset of at least $kn/d$ examples $S_V := S \cap V \subseteq S$ that lies in some $k$-dimensional subspace $V$ and some invertible matrix $A$ such that $F_A(S_V)$ is in $1/2k$-approximate radially isotropic position (up to isomorphic to $\mathbb{R}^k$). Now, for convenience, we assume our transformed data $F_A(S_V)$ is exactly our original dataset and we focus on the transformed data. Notice that for each $x \in S_V$, we have

$$\mathrm{sign}(w^* \cdot x) = \mathrm{sign}(A^{-T}w^* \cdot Ax) = \mathrm{sign}(A^{-T}w^* \cdot F_A(x)) = \mathrm{sign}(\mathrm{proj}_{A(V)}(A^{-T}w^*) \cdot f_A(x)),$$

which implies that each transformed example $F_A(x)$ is labeled by halfspace $v^* = \mathrm{proj}_{A(V)}(A^{-T}w^*)$ and has the same label as $x$. So, we can use Algorithm 1 to learn their labels. However, as Algorithm 1 is run over the transformed data, we have to simulate every TSQ used by the algorithm via a TSQ over the original data. This issue can be overcome using the following argument. Since $F_A$ is a bijection between $x$ and $F_A(x)$ and the outcome of the function $\phi(x, y)$ used in a TSQ for each example $x$ can be uniquely represented by two numbers, we can rewrite $\phi(F_A(x), y)$ as a function of $x$ for each $F_A(x)$ such that for a TSQ as long as $y(F_A(x)) = f(x)$, the result of the query will be the same. This gives us a way to simulate the TSQ over $S$.

## 3 Agnostic Learning with Threshold SQ

In this section, we study learning with TSQs under the more challenging adversarial label noise proving Theorem 1.5. In the previous section, we saw that using TSQ, learning halfspaces only requires $\mathrm{polylog}(1/\epsilon)$ rounds of interactions. We show in this section that this is not the case for the adversarial label noise. We show that it is impossible to reduce the query complexity from $\mathrm{poly}(1/\epsilon)$ to $\mathrm{polylog}(1/\epsilon)$ even for very simple classes such as the class of singletons (and thus the class of the halfspace in high dimensions). The classic method of proving query complexity lower bound [15, 16, 28] is to construct a hard instance directly. However, as there are infinite types of TSQs to be considered, it is impossible to construct a single hard instance that defeats all types of TSQs. Instead, we will build a reduction from a hardness result on agnostic distributed learning [32] that we define as follows.

**Definition 3.1** (Agnostic Distributed Learning)**.** *Let $\mathcal{X}$ be the space of examples. Let $a, b$ be two learners and $S =< S_a, S_b >$ be a collection of labeled examples, where $S_a$ is the (multi)set of labeled examples owned by $a$ and $S_b$ is the (multi)set of labeled examples owned by $b$. $a, b$ only knows their own sample set. A learning protocol is a communication strategy, where in each round of*

*communication $a$ sends information by bits to $b$ and after reviving information sent from $a$, $b$ sends information by bits back to $a$ and finally the learning protocol outputs a hypothesis $\hat{f} : \mathcal{X} \to \{\pm 1\}$. The error of $\hat{f}$ is*

$$\text{err}(\hat{f}) := \frac{1}{|S|} \sum_{x \in S} \mathbf{1}(\hat{f}(x) \neq f(x)),$$

*where $f(x)$ is the true label of $x$. Let $S$ be a collection of labeled examples, and $H$ be a hypothesis class. Given an accurate parameter $\epsilon \in (0, 1)$, the goal of the agnostic distributed learning problem is to design a learning protocol that outputs some $\hat{f}$ such that $\text{err}(\hat{f}) \leq \min_{h \in H} \text{err}(h) + \epsilon$ while minimizing the number of bits communicated in the learning protocol.*

In this paper, we will make use of the following slightly easier problem of agnostic distributed learning singleton functions, where the unlabeled examples owned by $a, b$ are known to each other and they want to output a labeling with error at most opt.

**Problem 3.2** (Distributed Learning Singleton). *Consider the agnostic distributed problem. Let $S = < S_a, S_b >$ be a collection of examples, where for $u \in \{a, b\}$, $S_u = \{(i, y_u^i)\}_{i=1}^n$, where $y_i^u \in \{\pm 1\}$ for $i \in [n]$. Let $H = \{h_i(x) = 2\mathbf{1}(x = i) - 1 \mid i \in \mathbb{N}\}$ be the class of singleton functions. The goal is to design a (randomized) learning protocol that outputs a hypothesis $\hat{f}$ such that $\text{err}(\hat{f}) \leq \min_{h \in H} \text{err}(h) + \epsilon$ for $\epsilon = 1/4n$ with probability at least $2/3$.*

[32] shows the following hardness result for Problem 3.2.

**Theorem 3.3** (Lemma 3 in [32]). *If there is a (randomized) learning protocol that can solve Problem 3.2 using $T(n)$ bits of communication, then there is a (randomized) protocol that can solve the set-disjointness problem with $T(n) \log(n)$ bits of communication.*

According to [29], solving the set disjointness problem requires $\Omega(n)$ bits of communication, and thus solving Problem 3.2 requires $\tilde{\Omega}(n) = \tilde{\Omega}(1/\epsilon)$ bits of communication. The central result we use to prove Theorem 1.5 is the following technical lemma, which means if one can agnostically learn the class of singleton functions with error opt using $T(n)$ queries, then one can design a learning protocol for Problem 3.2 with $T(n)\text{polylog}(n)$ bits of communication. This is enough to prove Theorem 1.5, because given the hardness of Lemma 3.4, we can create a hard problem by making multiple copies of each example used in the proof of Lemma 3.4. This preserves the error of every hypothesis $h : \mathcal{X} \to \{\pm 1\}$. We leave more details to Appendix B and in the rest of this section, we give an overview of the proof of Lemma 3.4.

**Lemma 3.4.** *Let $S \subseteq \mathbb{N}$ be a multiset of $2n$ examples and $f(x)$ be a hidden labeling function. Let $\mathcal{H} = \{h_i(x) = 2\mathbf{1}(x = i) - 1 \mid i \in \mathbb{N}\}$ be the class of singleton functions. If there is an algorithm $\mathcal{A}$ that can make $T(n)$ TSQ and outputs some $\hat{f}$ such that $\text{err}(\hat{f}) \leq \min_{h \in H} \text{err}(h) + \epsilon$, with $\epsilon = 1/4n$, then there is a learning protocol that can solve Problem 3.2 with $O(T(n)\text{polylog}(n))$ bits of communications.*

Consider $\mathcal{A}$ to be a learning algorithm for singleton functions that can learn up to error opt with $T(n)$ queries. Since both $a, b$ know the unlabeled examples owned by each other and know the labels of examples owned by themselves, we will design a learning protocol for $a, b$ to check the answer to each TSQ used by $\mathcal{A}$ together using only $\text{polylog}(n)$ bits of communication. Recall that in the definition of TSQ, each $q_i$ answers if $\sum_{x \in S} \phi(x, y) \geq \tau$, where $\phi(x, y)$ given every $x$ is a two-value function. Thus, to check the answer of $q_i$, it is sufficient to check if $\sum_{x \in S_a} \phi(x, y) \geq \tau - \sum_{x \in S_b} \phi(x, y)$. One possible way to check the answer is to let $a$ send the number $\sum_{x \in S_a} \phi(x, y)$ to $b$. However, if a TSQ is very complicated, communicating such a number would cost too many bits. Two arguments are made to address this problem. First, we claim that we can assume every outcome of $\sum_{x \in S_a} \phi(x, y)$ and $\tau - \sum_{x \in S_b} \phi(x, y)$ is an integer with bit complexity $n$. Intuitively, this is because there are at most $2^n$ different outcomes for $\sum_{x \in S_a} \phi(x, y)$ and $\tau - \sum_{x \in S_b} \phi(x, y)$ and we can explicitly create a map from each outcome to such an integer. Second, we show that to compare a pair of integers with bit complexity $n$ only $\text{polylog}(n)$ bits of communication are required. To see why this is true, we can expand integers $I_a = \sum_{x \in S_a} \phi(x, y)$ and $I_b = \tau - \sum_{x \in S_b} \phi(x, y)$ into binary strings. Then $I_a > I_b$ if and only if there exists some index $i$ such that $(I_a)_j = (I_b)_j$ for each $j > i$ but $(I_a)_j > (I_b)_j$ for each $j = i$. Thus, to compare $I_a, I_b$, we only need to binary search the first index $j$ such that the

partial binary strings of $I_a, I_b$ are different. Since checking whether two binary strings are equal only requires $O(\log n)$ bits of communication, we only need $O(\log^2 n)$ bits of communication to compare the two integers.

## Acknowledgments

This work was supported by the NSF Award CCF-2144298 (CAREER).

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

# Supplementary Material

## A  Omitted Proofs and Discussions in Section 2

### A.1  Proof of Theorem 2.1

*Proof of Theorem 2.1.* Since $u \cdot w^* \geq 1/(4\sqrt{k})$, we know that $w^* \in P_0$, furthermore, $K := B^k_{\epsilon/\mathrm{poly}(k)}(w^*) \cap P_0$ contains a ball in $V$ with radius at least $\epsilon/\mathrm{poly}(k)$. In particular, $0 \notin P_0$. We will first show that every time Algorithm 1 computes $((c_i - u/4)^t, 0)$, it separates $x_i$ and $K$.

For a given round $i$ in Algorithm 1, for every $x \in S$, $Y_x = 1$ implies that $w_i$ misclassifies $x$. According to Algorithm 1, we know that when $((c_i - u/4)^t, 0)$ is computed, it must be the case where $\mathbf{E}_{x \sim S_{w_i}} Y_x \geq \eta + \epsilon$. We remark that we can use a single TSQ to check if $\mathbf{E}_{x \sim S_{w_i}} Y_x \geq \eta + \epsilon$ by querying if the number of mistakes made by $w_i$ over $S_{w_i}$ is larger than $(\eta + \epsilon)|S_{w_i}|$.

Since $U$ is a random subset of $S_{w_i}$ with size $m = \tilde{O}(k^2/\epsilon^2)$, by Hoeffding's inequality we know that with probability at least $1 - \mathrm{poly}(\delta)$, $\frac{1}{m} \sum_{x \in U} Y_x \geq \eta + \epsilon/2$. We first show that given this happens, $\bar{c}_i := \frac{1}{m} \sum_{x \in U} \phi(x, y)x$ must have large correlation with $w_i$. We have

$$\bar{c}_i \cdot w_i := \frac{1}{m} \sum_{x \in U} \phi(x, y)x \cdot w_i = \frac{1}{m} \sum_{x \in U} \frac{(Y_x(y) - \eta)}{(w_i \cdot x)}(x \cdot w_i) = \frac{1}{m} \sum_{x \in U} (Y_x(y) - \eta) \geq \epsilon/2,$$

where in the last inequality we use the fat that $\frac{1}{m} \sum_{x \in U} Y_x \geq \eta + \epsilon/2$. On the other hand, we show that with high probability, $\frac{1}{m} \sum_{x \in U} \phi(x, y)x \cdot w^* \leq \epsilon/(20\sqrt{k})$. To see this, we first consider any fixed $x \in S$. If $\mathrm{sign}(w_i \cdot x) = \mathrm{sign}(w^* \cdot x)$, then under the Massart noise model, in expectation we have

$$\mathbf{E}_{y(x)} \phi(x, y)x \cdot w^* = \mathbf{E}_{y(x)} \frac{Y_x(y) - \eta}{(w_i \cdot x)}(w^* \cdot x) = \frac{\eta(x) - \eta}{(w_i \cdot x)}(w^* \cdot x) \leq 0.$$

Similarly, if $\mathrm{sign}(w_i \cdot x) \neq \mathrm{sign}(w^* \cdot x)$, then

$$\mathbf{E}_{y(x)} \phi(x, y)x \cdot w^* = \mathbf{E}_{y(x)} \frac{Y_x(y) - \eta}{(w_i \cdot x)}(w^* \cdot x) = \frac{1 - \eta(x) - \eta}{(w_i \cdot x)}(w^* \cdot x) \leq 0.$$

Thus, for any possible subset $U \subseteq S$, it always holds that $\frac{1}{|U|\epsilon} \sum_{x \in U} \mathbf{E}_{y(x)} \phi(x, y)x \cdot w^* \leq 0$, where the randomness comes from the Massart noise model. In Algorithm 1, each $x \in U$ satisfies $|w_i \cdot x| \geq 1/(4\sqrt{k})$. This implies that it always holds for each $x \in U$ that $\phi(x, y)x \cdot w^*/\epsilon \in (-4\sqrt{k}/\epsilon, 4\sqrt{k}/\epsilon)$. Since $U$ is a random subset of $k^2/\epsilon^2$ examples from $S_{w_i}$, by Hoeffding's inequality, we have

$$\mathbf{Pr}\left(\frac{1}{m\epsilon} \sum_{x \in U} \phi(x, y)x \cdot w^* - \frac{1}{m\epsilon} \sum_{x \in U} \mathbf{E}_{y(x)} \phi(x, y)x \cdot w^* \geq \frac{1}{20\sqrt{k}}\right) \leq \exp(-\frac{\epsilon^2 m}{k^2}) \leq 1 - \mathrm{poly}(\delta).$$

Thus, with high probability $\bar{c}_i \cdot w^* \leq \epsilon/(20\sqrt{k})$.

Notice that for each coordinate $j$, $|\bar{c}_{ij}| \leq 4\sqrt{k}$. Along each coordinate $j$, we are able to use TSQ of the type $\frac{1}{|U|\epsilon} \sum_{x \in U} \phi(x, y)(x)_j \geq \tau$ to binary search $\bar{c}_{ij}$ up to error $\epsilon/(8k^2)$ in $O(\log(k/\epsilon)) \leq O(\log(d/\epsilon))$ rounds of interactions. Since we have found $\|c_i - \bar{c}_i\|_\infty \leq \epsilon/(8k^2)$, we know that

$$c_i \cdot w_i \geq \bar{c}_i \cdot w_i - \epsilon/(8k) \geq \epsilon/2 - \epsilon/(8k) \geq 3\epsilon/8$$
$$c_i \cdot w^* \leq \bar{c}_i \cdot w^* + \epsilon/(8k) \leq \epsilon/(20\sqrt{k}) + \epsilon/(8k) \leq \epsilon/(17\sqrt{k}).$$

However, $c_i$ itself cannot separate $x_i$ from $K$ as it could be the case that both $c_i \cdot w^*$ and $c_i \cdot w_i$ are positive. However, since $u \cdot w^* \geq 1/(4\sqrt{k})$ and $\|u\|_2 = 1$, $c_i - \epsilon u/4$ can separate $w_i$ from $K$. This can be viewed as follows. On the one hand,

$$(c_i - \epsilon u/4) \cdot w_i \geq c_i \cdot w_i - \epsilon/4 \geq \epsilon/8 > 0,$$

which means $(c_i - u\epsilon/4) \cdot x_i > 0$. On the other hand, for every $x \in K$, we have

$$(c_i - \epsilon u/4) \cdot x \leq (c_i - \epsilon u/4) \cdot w^* + \epsilon/\mathrm{poly}(k) \leq \epsilon/(17\sqrt{k}) - \epsilon/(16\sqrt{k}) + \epsilon/\mathrm{poly}(k) < 0.$$

As long as Algorithm 1 computes $(c_i - \epsilon u/4)$, with high probability it will separate $x_i$ from $K$. In particular, by Theorem 2.3, we know that after $T = \tilde{O}(k \log(1/\epsilon))$ rounds, any point in $P_T$ must be at most $\epsilon/\text{poly}(\log(1/\epsilon))$ close to $w^*$. This implies that over $S_{w_T} = \{x \in S \,|\, w_T| \cdot x \geq 1/(2\sqrt{k})\}$, $w_T$ and $w^*$ agrees on every single example in $S_{w_T}$. Thus, after $\tilde{O}(k \log(1/\epsilon))$ rounds, Algorithm 1 is guaranteed to output some $w_i$ such that $w_i$ has an error at most $\eta + \epsilon$ over the region $S_{w_i}$. By our assumption, $S_{w_i}$ has a size at least $n/(4k)$. This proves the correctness of the algorithm.

Finally, we compute the query complexity of the algorithm. In each round of Algorithm 1, we use 1 TSQ to check if the current hypothesis is good enough and use $\tilde{O}(d \log(1/\epsilon))$ TSQ to find a good approximation of the separation hyperplane. Since there are at most $\tilde{O}(k \log(1/\epsilon))$ rounds, the query complexity of Algorithm 1 is $\tilde{O}(d^2 \log^2(1/\epsilon))$

$\square$

## A.2 Proof of Theorem 1.4

*Proof of Theorem 1.4.* We start by showing the correctness of Algorithm 2. We will show that in each round of Algorithm 2, $|S'| \geq |S|/d$ and $\hat{f}$ over $S$ has error at most $\eta + \epsilon/2$. Given this to be correct, after at most $O(d \log(1/\epsilon))$ rounds, Algorithm 2 labels $(1 - \epsilon/2)$ fraction of the examples with an error of $\eta + \epsilon/2$, leaving at most $\epsilon/2$ fraction of the examples unlabeled. This means $\hat{f}$ has error at most $\eta + \epsilon$.

By Lemma 2.5 and Theorem 2.6, we know that in each round of Algorithm 2, we can compute we find a subspace $V$ that contains $k/d$ fraction of the unlabeled data in $S$ and a matrix $A$ that can make $F_A(S \cap V)$ in approximate radially isotropic position. If $w^* \perp V$, then the ground truth labels of examples in $S \cap V$ are the same and thus with high probability a constant hypothesis achieves an error of at most $\eta + \epsilon$ over $S \cap V$.

Now we assume $w^*$ is not orthogonal to $V$ and we will show that by running Algorithm 1 $\tilde{O}(\log(1/\delta))$ times, we are able to label $S' \subseteq S \cap V$, a subset of at least $1/k$-fraction of examples in $S \cap V$ with error at most $\eta + \epsilon$. To see this, we first argue that labeling $S \cap V$ is equivalent to labeling the transformed data $F_A(S \cap V)$. We notice that for every $x \in V$, we have

$$\text{sign}(w^* \cdot x) = \text{sign}(A^{-T} w^* \cdot Ax) = \text{sign}(A^{-T} w^* \cdot F_A(x)) = \text{sign}(\text{proj}_{A(V)}(A^{-T} w^*) \cdot F_A(x)),$$

which implies that we can view $F_A(V)$ to be labeled by a halfspace $v^* = \text{proj}_{A(V)}(A^{-T} w^*)$ furthermore, $x$ and $F_A(x)$ have the same ground truth label. If we associate $y(F_A(x)) = f(x)$ for each $x \in V$, then the label $y(F_A(x))$ of $F_A(x)$ can be seen as created by halfspace $\text{sign}(v^* \cdot x)$ under the Massart noise model such that $\eta(F_A(x)) = \eta(x), \forall x$. Thus, if we are able to find a subset $F_A(S' \cap V) \subseteq F_A(S \cap V)$ and label examples in $F_A(S' \cap V)$ up to error $\eta + \epsilon/2$, then we are able to label the labels of the corresponding examples in $S'$ up to error $\eta + \epsilon/2$. We will use Algorithm 1 to do this. Since $F_A(S \cap V)$ are in approximate radially isotropic position, we know from that Lemma 2.5 that for every unit vector $w \in V$, at least $1/(4k)$ fraction of examples in $F_A(S \cap V)$ satisfied $|F_A(x) \cdot w| \geq 1/(2\sqrt{k})$. Thus, once Algorithm 1 outputs a labeling $\hat{f}_{S'}$ for $S' \subseteq F_A(S \cap V)$, the size of $S'$ is at least $|F_A(S \cap V)|/(4k) \geq |S|/(4d)$ and the error of $\hat{f}_{S'}$ must be at most $\eta + \epsilon/2$. By Theorem 2.1, we need some unit vector $u$ that has a non-trivial correlation with the target halfspace $A^{-T} w^*$ for the transformed data. By randomly select a unit vector in $V$, with constant probability (see [45]), we can guarantee that $u \cdot \text{proj}_{A(V)}(A^{-T} w^*) \geq 1/(4\sqrt{k})$. Thus by repeating Algorithm 1 several times, with high probability, it will output some labeling function. However, to run Algorithm 1, we have to implement TSQ over the transformed data while we can only make TSQ over the original data. Such an issue is easy to address. Since $F_A$ is a bijection between $x$ and $F_A(x)$ and the outcome of the function $\phi(x, y)$ for each example $x$ can be uniquely represented by two numbers, we can rewrite $\phi(F_A(x), y)$ as a function of $x$ for each $F_A(x)$ such that for a TSQ as long as $y(F_A(x)) = f(x)$, the result of the query will be the same. As we have mentioned that $y(F_A(x)) = f(x)$ holds for every example, we conclude that we can simulate each TSQ over the transformed data via a TSQ over the original data. This finishes the proof of the correctness of Algorithm 2.

Finally, we calculate the query complexity of Algorithm 2. By Theorem 2.1, we know that every time we run Algorithm 1, we make $\tilde{O}(d^2 \log^2(1/\epsilon))$ queries. Since every round of Algorithm 2, we run

Algorithm 2 $O(\log(1/\epsilon))$ rounds and there are at most $O(d \log(1/\epsilon))$ rounds, we conclude the query complexity of Algorithm 2 is $O(d^3 \log^3(1/\epsilon))$. In particular, by Theorem 2.6 and Algorithm 1, each subroutine of the algorithm can be implemented in polynomial time, we conclude that Algorithm 2 can be run in polynomial time.

$\square$

### A.3    On the TSQs Used by Algorithm 2

In this section, we want to discuss the TSQs used by Algorithm 2 and argue that these TSQs have simple structures and are easy to communicate and implement. There are two types of TSQs used by Algorithm 2.

First, the algorithm needs to check whether a hypothesis $h = \text{sign}(w \cdot x)$ has an error larger than $\tau$ over a given region $U$. In other words, we want to use TSQs to approximate the conditional expectation, $\mathbf{E}_x \mathbf{1}(\text{sign}(w \cdot x) \neq f(x) \mid x \in U)$. To express this using TSQ, for each $x \in U$, we define $\phi(x, y) = 1/|U|$ if $\text{sign}(w \cdot x) \neq y$ and 0 otherwise. For each $x \in S \setminus U$, we define $\phi(x, y) = 0$. In particular, in Algorithm 2 each $U$ we use is just some random samples drawn from $S \cap \{x \in V \mid |F_A(x) \cdot w| \geq \Omega(1/\sqrt{d})\}$. To communicate such a TSQ, a learner only needs to communicate such a query, a learner only needs to communicate to the labeler, $(v, A)$, the parameters for a Forster's transformation, $w$, the hypothesis maintained by the algorithm, $\tau$, the threshold used by the TSQ and a random seed to guide the labeler to do sampling. The labeler receives these parameters, computes the answer to the TSQ, and returns a binary answer to the learner.

The second type of TSQ can be seen as a weighted sum of the mistakes made by the current hypothesis $h$ over a region $U$. Recall the notation used in Algorithm 1, $(Y_x(y) - \eta)/(w_i \cdot x)$, where $Y_x(y) = 1$ if $h$ makes a mistake at $x$. The algorithm wants to approximate the point $\mathbf{E}_x(F_A(x)(Y_x(y) - \eta)/(w_i \cdot F_A(x)) \mid x \in U)$, which is equivalent to get an approximation of the point from each coordinate. Similarly, every $U$ used by the algorithm is a random set sample from $S \cap \{x \in V \mid |F_A(x) \cdot w| \geq \Omega(1/\sqrt{d})\}$. To communicate such a query, a learner will communicate $(v, A)$, the parameters for a Forster's transformation, $w$, the hypothesis maintained by the algorithm, $i$, the coordinate the learner want to approximate, $\tau$, the threshold used by the TSQ and a random seed to guide the labeler to do sampling.

This shows that the TSQs used by Algorithm 2 is simple from both a computational view and a communication complexity view.

## B    Omitted Proofs in Section 3

### B.1    Proof of Lemma 3.4

*Proof of Lemma 3.4.* Notice that the learning algorithm $\mathcal{A}$ can be described as follows. In round $i$, $\mathcal{A}$ constructs a TSQ $q_i$ (possibly using randomness), submits $q_i$ to the labeler, and receives the answer to $q_i$. Given $\mathcal{A}$, we will design a learning protocol as follows. In round $i$, the learner $a, b$ will check the answer of $q_i$ together by sending bits to each other and construct the next TSQ $q_{i+1}$ based on the answer and using $\mathcal{A}$. If they use only $K(n)$ bits of communication to check $q_i$ in each round, then since $\mathcal{A}$ will output some $\hat{h}$ such that $\text{err}(\hat{f}) \leq \min_{h \in H} \text{err}(h) + \epsilon$ after $T(n)$ rounds, the total bits of communication is at most $T(n)K(n)$.

We can without loss of generality assume the randomness used to implement $\mathcal{A}$ is public so that in each round, both $a$ and $b$ know exactly the TSQ $q_i$. Otherwise, by Newman's theorem [36], we only need to use another $O(\log n)$ bits of communication to simulate the randomness used by $\mathcal{A}$. Recall that in the definition of TSQ, each $q_i$ answers if $\sum_{x \in S} \phi(x, y) \geq \tau$, where $\phi(x, y)$ given every $x$ is a two-value function. Thus, to check the answer of $q_i$, it is sufficient to check if $\sum_{x \in S_a} \phi(x, y) \geq \tau - \sum_{x \in S_b} \phi(x, y)$. Although $a$ can compute $\sum_{x \in S_a} \phi(x, y)$ and sends the number to $b$, communicating a single number $\sum_{x \in S_a} \phi(x, y)$ might need a lot of bits of communication. In the rest of the proof, we will design a protocol that use only $O(\log^2 n)$ bits of communication to check the answer of $q_i$.

First, we show that we can without loss of generality assume every outcome of $\sum_{x \in S_a} \phi(x, y)$ and $\tau - \sum_{x \in S_b} \phi(x, y)$ is an integer with bits complexity $n$. We observe that both $\sum_{x \in S_a} \phi(x, y)$ and $\tau - \sum_{x \in S_b} \phi(x, y)$ have at most $2^n$ outcomes. As $a$ and $b$ all know all possible outcomes, they can explicitly construct a maps $F_a$, which maps each of the possible outcomes of $\sum_{x \in S_a} \phi(x, y)$ to an integer between 0 and $2^n - 1$ and a map $F_b$, which maps each of the possible outcomes of $\tau - \sum_{x \in S_b} \phi(x, y)$ to an integer between 0 and $2^n - 1$. Given these two explicit maps, if $a$ and $b$ can use communication to learn $F_a(\sum_{x \in S_a} \phi(x, y))$ and $F_b(\tau - \sum_{x \in S_b} \phi(x, y))$, then they can reconstruct $\sum_{x \in S_a} \phi(x, y)$ and $\tau - \sum_{x \in S_b} \phi(x, y)$. In the rest of the proof, we prove based on this assumption.

Next, we design a protocol that uses $O(\log^2 n)$ bits of communication. After $a$ determine $\sum_{x \in S_a} \phi(x, y)$, $a$ know an integer $I_a$ such that $I_b = \tau - \sum_{x \in S_b} \phi(x, y) > \sum_{x \in S_a} \phi(x, y)$ if and only if $I_b > I_a$. So it remains to show that given two integers $I_a, I_b$ with bit complexity $n$, we are able to compare these two integers with $O(\log n)$ bits of communication. To do this, we first represent $I_a, I_b$ by binary strings of length $n$. Notice that $I_a > I_b$ if and only of there exists some index $i$ such that $(I_a)_j = (I_b)_j$ for $j > i$ but $(I_a)_j > (I_b)_j$ for $j = i$. This implies that to compare $I_a$ and $I_b$ it is sufficient to find the largest index $i^*$ such that $(I_a)_j = (I_b)_j$ for each $j > i^*$ and compare $(I_a)_{i^*}$ and $(I_b)_{i^*}$. Such an index can be found via binary search. Specifically, given $i$ we want to check if $(I_a)_j = (I_b)_j$ for each $j < i$. If the two partial binary strings are equal, then we decrease $i$, otherwise we increase $i$. After $O(\log n)$ rounds, we successfully find such an index $i^*$. It is well-known that checking the equality of two binary strings of length $n$ can be done via a simple randomized protocol by communicating $O(\log n)$ bits [36, 42]. Thus, in total, with $O(\log^2 n)$ bits of communication, we are able to compare $I_a, I_b$ and thus can check the answer of the TSQ.

This gives a randomized learning protocol that uses $O(T(n) \log^2(n))$ bits of communication. $\square$

### B.2 Proof of Theorem 1.5

*Proof of Theorem 1.5.* Let $\epsilon \in (0, 1)$. For simplicity, we write $\epsilon = 1/4n$ and let $S$ be a multiset of $2n$ examples over $\mathbb{N}$. Given any labeling function $f(x)$ over $S$ and every $m \geq 2n$ such that $m/(2n)$ is an integer, we create a multiset $S'$ of size $m$ in the following way. For each $x \in S$ we create $m/(2n)$ copies $x'$ for $x$ such that for each copy $x'$ it has a hidden label equal to $f(x)$. Denote by $f'$ the labeling function over $S'$. Notice that for every hypothesis $h : \mathbb{N} \to \{\pm 1\}$, the error of $h$ over $S'$ and the error of $h$ over $S$ are the same. This implies that if we have a learning algorithm such that for every $S' \subseteq \mathbb{N}$ and every labeling function $f'$, it can output a hypothesis $\hat{f}$ using $T(1/\epsilon) = T(4n)$ TSQs such that with probability $2/3$ $\hat{f}$ has an error opt $+ \epsilon$, then $\hat{f}$ has an error at most opt $+ \epsilon$ over the original dataset $S$. By Lemma 3.4, we know that this implies a learning protocol that solve Problem 3.2 with $O(T(1/\epsilon) \log^2(1/\epsilon)) = O(T(4n) \log^2(n))$ bits of communication. By Theorem 3.3, this implies a communication protocol that solves the set disjointness problem of size $n$ using $O(T(1/\epsilon) \log^3(1/\epsilon)) = O(T(4n) \log^3(n)))$ bits of communication. By [29], we know that to solve a set disjointness problem of size $n$, any (randomized) protocol has a communication complexity of $\Omega(n)$. This implies that $T(1/\epsilon) = \tilde{\Omega}(1/\epsilon)$.

$\square$

## C Proof of Theorem 1.6

In this section, we prove Theorem 1.6 by presenting the following Algorithm 3. Our algorithm is inspired by [5], where they design an algorithm that learns a hypothesis class $H$ with finite VC dimension up to error $O(\text{opt}) + \epsilon$ using $O(d \log(1/\epsilon))$ class-conditional queries, which returns an example with a specified label in a given region. Unlike their algorithm, our algorithm does not need such a strong query. Instead, our algorithm makes $O(d \log(1/\epsilon))$ TSQs to achieve the same guarantee. Furthermore, each TSQ used in Algorithm 3 only checks if a given hypothesis has an error larger than some threshold over a given region.

*Proof of Theorem 1.6.* If opt $\leq \epsilon$, the learning up to error $O(\text{opt}) + \epsilon$ is equivalent to learning up to error $O(\epsilon)$ and $\eta = \epsilon$ can be used as an upper bound for opt. So, we assume that $\eta = \text{opt} \geq \epsilon$, because we can always guess some $\eta$ such that $\eta/2 \leq \text{opt} \leq \eta$ via a doubling trick, which will only

---

**Algorithm 3** APPROXIMATE AGNOSTIC LEARNING(Learning a labeling up to $O(\text{opt})$ error)

---

**Input:** Dataset $S$ of size $n$, hypothesis class $H$ with VC dimension $d$, $\eta$, an upper bound of $\text{opt}$
**Output:** $\hat{f}$, a labeling of $S$
Let $H_0$ be an $\epsilon$-cover of the hypothesis class $H$ with respect to the uniform distribution over $S$
Let $f_{H_0}$ be a labeling over $S$. For each $x \in S$, $f_{H_0}(x)$ agrees with the majority of $H_0$ at $x$.
Use a single TSQ to check if the error $\eta_0$ of $f_{H_0}$ over $S$ is larger than $O(\eta)$
If $\eta_0 \leq O(\eta)$, output $f_{H_0}$
**while** the error $\eta_i$ of $f_{H_i}$ larger than $O(\eta)$ **do**          ▷ This can be checked with a single TSQ
    **for** $j = 1, \ldots, T = O(\log(1/\delta))$ **do**
        Keep drawing random subsets $S_j$ of size $1/(50\eta)$ from $S$ until $S_j$ gets accepted
        We accepted $S_i$ if we find at least 1 example in $S_j$ are misclassified by $f_{H_i}$ using a single
TSQ
        **if** More than $1/6$ fraction of the hypothesis in $H_i$ agrees with $f_{H_i}$ over $S_j$ **then**
            Mark all the hypothesis in $H_i$ that agrees with with $f_{H_i}$ over $S_j$
        **else**
            Find a subset of $\hat{S}_j \subseteq S_j$ such that $\xi$-fraction of the hypothesis in $H_i$ agrees with $f_{H_i}$
over $\hat{S}_j$, where $\xi \in [1/6, 2/3]$.
            Use a single TSQ to check if over $\hat{S}_j$, $f_{H_i}$ makes no mistake. If so, mark all hypotheses
that disagree with $f_{H_i}$ at any single example over $\hat{S}_j$, otherwise mark the hypothesis in $H_i$ agrees
with $f_{H_i}$ over $\hat{S}_j$
    Remove all hypotheses in $H_i$ that are marked more than $0.1T$ times and $H_{i+1}$ be the set of
remaining hypothesis
    **return** $f_{H_i}$

---

make the final guarantee worse up to a constant factor. Denote by $h^* \in H_0$ the hypothesis that has the smallest error over $S$. By the definition of $\epsilon$-cover, we know that $\text{err}(h^*) \leq \eta + \epsilon$.

We show that with high probability, in each round of Algorithm 3, either $\text{err}(h_{H_i})$ is at most $250\eta$ or a constant fraction of the hypothesis in $H_i$ gets removed. In particular, we will show that $h^*$ will always stay in $H_i$ and thus after $\tilde{O}(d\log(1/\epsilon))$ rounds, we are guaranteed to output some hypothesis with small error.

Assume that $\text{err}(h_{H_i}) > 250\eta$. We say a set $S_j$ is good if it contains no example $x$ such that $h^*(x) \neq f(x)$. We first show that given a set $S_j$ accepted by Appendix C, with a non-trivial probability it is good.

$$
\begin{aligned}
\mathbf{Pr}\left(S_j \text{ is good} \mid S_j \text{ is accepted}\right) &= \frac{\mathbf{Pr}\left(S_j \text{ is good and } S_j \text{ is accepted}\right)}{\mathbf{Pr}\left(S_j \text{ is accepted}\right)} \\
&\geq \mathbf{Pr}\left(S_j \text{ is good and } S_j \text{ is accepted}\right) \\
&\geq 1 - \mathbf{Pr}\left(S_j \text{ is not good}\right) - \mathbf{Pr}\left(S_j \text{ is not accepted}\right).
\end{aligned}
$$

Since the noise rate is $\eta$, we know from the definition of $\epsilon$-cover that $h^*$ has an error at most $2\eta$. This implies that in expectation, a random $S_J$ contains $1/25$ example that is misclassified by $h^*$.

$$
\mathbf{Pr}\left(S_j \text{ is not good}\right) = \mathbf{Pr}\left(S_j \text{ contains one example misclassified by} h^*\right) \leq 1/25 = 0.04.
$$

On the other hand, since $\text{err}(h_{H_i}) > 250\eta$, a random example has a probability at most $1 - 1/(250\eta)$ not misclassified by $h_{H_i}$ and this

$$
\mathbf{Pr}\left(S_j \text{ is not accepted}\right) \leq (1 - 1/(250\eta))^{1/(50\eta)} \leq e^{-5} \leq 0.01. \tag{2}
$$

Thus, with a probability of at least $95\%$, an accepted set $S_j$ is good. In particular, $h^*$ misclassified no example in $S_J$. This implies that $h^*$ will not get marked when $S_j$ is good. And thus in expectation, $h^*$ will not get marked for more than $T/20$ times. By Hoeffding's inequality, this implies with high probability $h^*$ will not get removed from $H_i$. On the other hand, for every $S_j$ that gets accepted more than $1/6$ of the hypothesized in $H_j$ must get marked. To show this, we consider two cases. In the first case, more than $1/6$ fraction of the hypothesis in $H_i$ agrees with $f_{H_i}$ over $S_j$. In this case, according to Algorithm 3, all hypothesizes in $H_i$ that agree with $f_{H_i}$ over $S_j$ will get marked and more than

1/6 fraction of the hypothesis in $H_i$ will be marked. In the second case, we show that there must be a subset of $\hat{S}_j \subseteq S_j$ such that $\xi$-fraction of the hypothesis in $H_i$ agrees with $f_{H_i}$ over $\hat{S}_j$, where $\xi \in [1/6, 2/3]$. We order $S_j$ in an arbitrary order $x_1, x_2, \ldots, x_m$, where $m = |S_j|$. For each $t \in [m]$, we use $H^{(t)}$ to denote the set of hypothesises in $H_i$ that agree with $h_{H_i}$ for $x_1, \ldots, x_t$. From the above discussion, we know that $|H^{(m)}| \le |H_i|/6$. On the other hand, we know by the definition of $h_{H_i}$ that $|H^{(1)}| \ge |H_i|/2$. If $|H^{(1)}| \le 2|H_i|/3$, then we are done. Otherwise, there must be a largest $t^*$ such that $|H^{(t^*)}| \ge 2|H_i|/3$. We claim that $|H_i|/6 \le |H^{(t^*+1)}| \le 2|H_i|/3$. This is because at most $|H_i|/2$ hypothesises in $H_i$ will disagree with $h_{H_i}$ over $x_{t+1}$ and will get deleted from $H^{(t^*)}$. This implies that we can choose $\hat{S}_j = \{x_1, \ldots, x_{t+1}\}$. Given this, whether $h_{H_j}$ makes a mistake over $\hat{S}_j$ or not, at least $1/6$ fraction of the hypothesis will be marked.

We next use this fact to show that in each round, a constant fraction of the hypotheses in $H_i$ will be removed. Assuming that $c$-fraction of the hypotheses in $H_i$ gets removed from $H_i$. On the one hand, for each accepted $S_j$, at least $|H_i|/6$ hypotheses are marked. So the total number of marks we made is at least $T|H_i|/6$. On the other hand, since only $c$-fraction of the hypotheses are marked by more than $0.1T$ times. The total number of marks we made is at most $c|H_i|T + 0.1(1-c)|H_i|T$. As the following inequality always holds

$$c|H_i|T + 0.1(1-c)|H_i|T \ge T|H_i|/6,$$

we conclude $c \ge 2/27$. According to [28], the size of the $\epsilon$-cover of $H$ is at most $O(d/\epsilon)^d$. Since $h^*$ is also included in $H_i$, after at most $k = \tilde{O}(d \log(1/\epsilon))$ rounds, $h^*$ will be the only hypothesis not removed. Since $h^*$ has an error at most $2\eta$, if Appendix C runs for $k$ rounds, then $h^*$ will be output. So, before the $k$th round, some $h_{H_i}$ must be output and has error $O(\eta) = O(\mathrm{opt})$. This proves the correctness of Algorithm 3.

Finally, it remains to prove the query complexity of Algorithm 3. We notice by Equation (2) that when $h_{H_i}$ has an error larger than $250\eta$, a random $S_j$ has only probability less than $0.01$ not getting accepted. This implies that to get an accepted $S_j$, we only need to make $\tilde{O}(1)$ TSQs to check whether $h_{H_i}$ has zero error over $S_j$. Since checking the error of $h_{H_j}$ and marking hypothesizes after some $S_j$ gets accepted will only take us $O(1)$ TSQs. In each round of Algorithm 3, we will make at most $\tilde{O}(1)$ TSQs. Since there are at most $\tilde{O}(d \log(1/\epsilon))$ rounds in Algorithm 3, we conclude the query complexity of Algorithm 3 is $\tilde{O}(d \log(1/\epsilon))$.

$\square$

