# OpenReview forum: "Active Classification with Few Queries under Misspecification"
_NeurIPS.cc/2024/Conference — NeurIPS 2024 spotlight_

### Official Review · Reviewer_DNzP · 2024-07-08

**Soundness:** 4
**Presentation:** 3
**Contribution:** 3
**Rating:** 7
**Confidence:** 3

**Summary:**

This paper considers active learning of halfspaces with noise that is both computationally efficient and query efficient without distributional assumption on X. Since it is known such problem is "hard" in the standard label query paradigm, it considers a new query model called "threshold statistical queries" (TSQ) where given a function $\phi$, set $S=\{ x_1, .., x_n \} $, and threshold $\tau$, the oracle answers if $\sum_{x_i \in S}\phi(x_i, y_i) > \tau$. Under Massart noise, it gives an active learning algorithm that is both computationally efficient and query efficient with this query model. On the other hand, it shows that under adversarial noise, this noise model cannot lead to query efficient learning.

**Strengths:**

- This paper considers a niche but still quite relevant theory problem of active learning for halfspaces. Its results are interesting: it shows that the newly proposed TSQ query model allows for efficient learning of halfspaces under Massart noise without distributional assumptions, while it is not strong enough to resolve the problem under the adversarial noise.

- The proposed TSQ query model is a nice and novel generalization/modification of previous models (statistical queries, region queries). The techniques for both the upper bound and lower bound look non-trivial and novel to me.

- The paper is written clearly. It provides a comprehensive review of related work and techniques, and clear high-level intuition behind the main results.

- The paper is sound, though I did not check proofs in Appendix.

**Weaknesses:**

- The label complexity bound in Theorem 1.4 is cubic in both d and log(1/epsilon). It would be interesting to see if this can be improved, but this would be a very minor issue.

- It would be more interesting if the authors could comment on how/whether the TSQ query model can make improvement in other noise models, or problems with a general (beyond linear) learning space.

**Questions:**

N/A

---

> ### Author Rebuttal · Authors · 2024-08-07
>
> We thank the reviewer for appreciating our work. Improving the cubic term in the label complexity is definitely an interesting open question. We will add a conclusion section in the future version of the work to introduce some potential future directions.

---

> > ### Comment · Reviewer_DNzP · 2024-08-12
> >
> > Thanks for the response. I will keep my score and support its acceptance.

---

### Official Review · Reviewer_WAF8 · 2024-07-09

**Soundness:** 4
**Presentation:** 3
**Contribution:** 3
**Rating:** 7
**Confidence:** 3

**Summary:**

This paper focused on the problem of active learning from enriched queries.
In order to be abale to learn halfspaces without restrictive distribution assumptions, authors propose Threshold Statistical Query (TSQ) which genarizes the region query and the classic statistical learning model query.
Using the proposed from of query, authors propose an algorithm to learn halfspaces in polynomial time, using only poly log(1/$\epsilon$) TSQ queries, under random classifcaiton noise and massart noise. Authors also proved the impossibility for the advasarial noise case.

**Strengths:**

- The proposed query has motivations easy to understand, and successfully derives the learning algorithm with carefully designed steps.
- The paper is self-containment. Essential background and definitions are clearly presented.
- The discussion on algorithm design and theory deriviation is solid and clear.

**Weaknesses:**

- The manusciprt seems to be not complete. It is not proper concluded.
  - An overal dicussion section on limitations may also improve the manuscript.
- The presentation of algorithm design can be improved.
  - Although already demonstrated in section 2.2, intuitive introduction on how the strong learning algorithm is designed and how it uses the weak learning algorithm can be added at the beginning of section 2.

**Questions:**

- On the noise format, authors considers the persistent form of noise which keeps same among different query trials.
  - Is this the reason that existing methods cannot achieve the poly log(1/$\epsilon$) query complexity?
  - How the proposed algorithm would be influenced if changing the noise to give random answers at each query? How will it influence finding $\bar{x}$?

**Limitations:**

The limitation on assumptions and adversarial noise is addressed in introduction.

---

> ### Author Rebuttal · Authors · 2024-08-07
>
> We appreciate the constructive feedback and thank the reviewer for carefully reviewing our manuscript. We will improve the presentation of the paper based on the suggestions and include a conclusion section with further discussion. Below is our response regarding the question on the noise format.
>
> >On the noise format, authors considers the persistent form of noise which keeps same among different query trials.
> Is this the reason that existing methods cannot achieve the $poly \log(1/\epsilon)$ query complexity?
> How the proposed algorithm would be influenced if changing the noise to give random answers at each query? How will it influence finding $x$?
>
> The same algorithm we developed works for the non-persistent setting where there is randomness in every query. This is because we construct queries using randomly selected examples. We will clarify this point in the manuscript.
> The reason that existing methods cannot achieve low query complexity is the noise and not whether it is persistent or not. As we mentioned in the introduction, a small amount of label noise could make every large region query useless as it would produce the same answer with high probability. The same would hold even if the noise is persistent. In fact, persistent noise is even more challenging as it prevents a learner from making repeated queries over a single example to figure out its underlying label, which in some settings could make the problem trivial.

---

> > ### Comment · Reviewer_WAF8 · 2024-08-09
> >
> > I thank author clearly address my question on the reason of infeasibility of existing methods.
> > I also read other reviewers comments with more knowledge of the field and corresonponding author response.
> > I would like to keep my score.

---

### Official Review · Reviewer_vqip · 2024-07-16

**Soundness:** 4
**Presentation:** 3
**Contribution:** 3
**Rating:** 7
**Confidence:** 3

**Summary:**

This paper extends the work on pool-based active learning from the realizable case to non-realizable settings: where the observed labels (or answers to active learning queries) can have noise. The paper focuses on learning half-spaces, a fundamental learning theory problem.

Existing works on pool-based active learning have designed several query languages that enable learning half-spaces up to $\epsilon$ accuracy using $O(\log(1/\epsilon))$ queries. The algorithms in these works, however, break when there is noise in the responses – even due to benign noise models such as random classification noise.

This work designs a new query language – threshold statistical queries – and shows that it enables learning of half-spaces up to $\epsilon$ accuracy with $\mathrm{poly}\log(1/\epsilon)$ samples even when the underlying labels have been corrupted by certain the Massart noise model (which is a significant extension of the random classification noise model).

To complement this result, the authors show that threshold statistical queries are not sufficient to learn halfspaces (or even 1-d thresholds) up to $\epsilon$ accuracy with fewer than $O(1/\epsilon)$ samples in the stronger agnostic learning model.

Finally, if instead of the usual $OPT+\epsilon$ guarantee, one considers algorithms with $O(OPT)+\epsilon$ guarantees, then the authors show that it is possible to learn halfspaces with $\tilde{O}(\log(1/\epsilon))$ samples.

**Strengths:**

Designing new types of queries that enable learning with exponentially fewer samples than necessary for PAC learning is an important area of active work. This paper studies a central problem in this area: learning halfspaces. As far as I understand, this is the first work to propose a type of query that enables learning halfspaces up to accuracy $\epsilon$ with $\mathrm{poly}\log(1/\epsilon)$ samples when there can be some noise in the answers. The types of noise models considered are also natural and widely studied in standard PAC learning settings.

While I am familiar enough with this area, to comment on the novelty of the techniques, the approach and the tools used (e.g., Forster’s transform) seem natural.

Something that can potentially be an added strength is if there is hope to use threshold statistical queries (introduced in this work) for learning other hypothesis classes, even non-efficiently: for instance, instance intersections of halfspaces or (non-axis-aligned) boxes. It would be great to have some discussion on this.

**Weaknesses:**

One weakness of the paper is the presentation: while the paper was largely clear and not hard to follow, some sentences and notation are a bit weird, but I am sure the paper would read much better after another pass.

Some specific notation-related suggestions/comments:
1. The notation B_1^k used in Theorem 2.1 is non-standard but is used before it is defined. Perhaps, till this notation is introduced, it can be avoided or explained in a footnote.
2. Definition 1.1 is a bit confusing. Specifically the sentence “Each query q : 2S×{±1} → {0, 1} …number in {0, 1}.” I am not sure what “given unknown labels as input” means.
3. I think using \cdot for the inner product is non-standard. Maybe it is better to use the $x^\top y$ notation?
4. Definition 3.1 and some subsequent places use the notation “S =< Sa, Sb >” to denote a set $\{S_a,S_b\}. Perhaps using braces is better?

**Questions:**

**Question 1.** The related works section mentions that Theorem 1.4 (the main/first result) can be implemented with standard statistical queries, as opposed to the new, threshold statistical queries introduced in this work. This is a bit confusing to me: if statistical queries are sufficient, then why is the model of threshold statistical queries introduced for Theorems 1.4 and 1.5 – the first two results? (Since Theorem 1.5 shows an impossibility result for learning with $\mathrm{poly}\log(1/\epsilon)$ threshold statistical queries and, further, since statistical queries are a special case of threshold statistical queries, this result should also hold for statistical queries.)

If both of these results do indeed work with statistical queries, then I think
1. this should be emphasized and
2. the definition of threshold statistical queries can be delayed till the last result.

**Question 2.** The results in this paper work for origin-centered half-spaces, instead of all halfspaces. Is this correct? I do not think this is a major concern as studying origin-centered halfspaces is a standard first step toward developing algorithms for half-spaces. But I think, if my claim is correct, then this should be clarified early in the introduction.

**Limitations:**

Please see my comments in weakness and questions sections.

---

> ### Author Rebuttal · Authors · 2024-08-07
>
> We thank the reviewer for appreciating our work and providing many constructive suggestions. We will improve the presentation in the future version of the manuscript. We think exploring the power of TSQ for other hypothesis classes such as the intersection of halfspaces or high dimensional boxes can be very promising directions and we will add more discussion on related directions in the future version of the manuscript. Below we respond to the comments and questions from the reviewer.
>
> >Question 1. The related works section mentions that Theorem 1.4 (the main/first result) can be implemented with standard statistical queries, as opposed to the new, threshold statistical queries introduced in this work. This is a bit confusing to me: if statistical queries are sufficient, then why is the model of threshold statistical queries introduced for Theorems 1.4 and 1.5 – the first two results? If both of these results do indeed work with statistical queries, then I think this should be emphasized and
> the definition of threshold statistical queries can be delayed till the last result.
>
> We want to remark that a high-level goal of this paper is to understand the power of queries for efficient learning with noise. Toward this goal, the query model studied should be simple but general enough to capture many existing query strategies. In this paper, we show that TSQ is an interesting general model that incorporates different query languages in the literature such as SQ with bounded precision, equivalence queries, region queries, label queries, etc.
>
> For Theorem 1.5, we want to mention that the return of a classic statistical query is a real number and the number of bits of information passed by such a query depends on the precision of the query. A statistical query with polynomial small precision can be implemented with a logarithmic number of TSQs and thus Theorem 1.5 also holds for SQs with bounded precision. However, as TSQ also captures other types of query languages that operate directly over samples, our lower bound in Theorem 1.5 shows that agnostic learnings are indeed intricate and to achieve a low query complexity, even more powerful queries are needed to come up with.
>
> On the other hand, TSQ is a very broad class of queries, and thus in some applications, the full generality of TSQs may not be feasible. An efficient learning algorithm using TSQs could in principle be optimized to use a subclass of TSQs with good structure that is easily implementable. For example, in Theorem 1.4, we use TSQs to find points $\bar{x}$ to feed Vaidya’s algorithm. As these queries are sampled from regions with non-trivial probability mass and estimating $\bar{x}$ up to a polynomial small error is enough, we can modify the algorithm by using the classic statistical queries with polynomial small precisions. On the other hand, in Theorem 1.6, the algorithm we use makes decisions based on both the statistical properties of the dataset and labels of individual examples, it cannot be implemented with solely SQ.
>
> We will expand these discussions and add more explanations together with examples to emphasize the relation between TSQ with other types of queries.
>
> >Question 2. The results in this paper work for origin-centered half-spaces, instead of all halfspaces. Is this correct?
>
> As our theorem is distributional-free(no distributional assumption is made), learning origin-centered halfspaces is equivalent to learning general halfspaces. This can be done by adding another dimension to the problem artificially.

---

> > ### Comment · Reviewer_vqip · 2024-08-11
> > **Response to authors**
> >
> > I thank the authors for taking the time to respond to my questions.
> >
> > I think it would be good to include some discussion comparing TSQ and SQ queries -- expanding upon what authors write in the rebuttal.
> >
> > Thanks for answering Question 2.
> >
> > I will retain my rating and think the paper should be accepted.

---

### Official Review · Reviewer_VrwT · 2024-07-23

**Soundness:** 4
**Presentation:** 4
**Contribution:** 4
**Rating:** 7
**Confidence:** 3

**Summary:**

This paper is concerned with active learning of halfspaces under persistent Massart noise, i.e., active learning under data $(X,Y)$ for which $\exists w^* \in \mathbb{R}^d, \eta \in [0,1/2)$ such that $P(Y = \mathrm{sign}(\langle w^*, X\rangle)) \ge 1-\eta$.

The paper proposes a new query language for this task, "threshold statistical queries" (TSQs). A TSQ consists of a set $S \subset (\mathcal{X}\times \mathcal{Y})^m,$ a map $\phi$ from $\mathcal{X} \times \mathcal{Y}$ to the reals, and a threshold, a real number, $\tau$, such that in response to a query of the form $(\phi, S,\tau),$ the learner receives a feedback indicating if $\sum_{(x_i, y_i) \in S} \phi(x_i, y_i) > \tau$ or not.

The main contribution of this work is the design and analysis of a new algorithm that active learns halfspaces in the described setting to error $\eta + \varepsilon$ with $O((d \log(1/\varepsilon)^3)$ TSQs and in polynomial time, without the use of structural assumptions on the feature distribution of the data. This method relies on an insightful way to use a small number of (simple) TSQs to both check if a vector $w$ approximates $\mathrm{sign}(\langle w^*, x\rangle)$ to $\eta + \varepsilon$-error _even under Massart noise_  over a set of $x$ that are isotropic and (near)-unit norm and satisfy $\{ |\langle w, x\rangle \ge 1/2\sqrt{\mathrm{dim}(x)} \},$ and if not, to build a witness for the same. This core dea is exploited to construct a convex feasibility oracle using a small number of TSQs via an existing method (Vaidya's Algorithm), which in turn is used to find an $\varepsilon$-neighbour of $w^*$ with a small number of TSQs and limited computation, at least over isotropic and near-unit-norm features. This is extended to general feature distributions by exploiting Forster's transform, as has appeared in recent work on learning halfspaces under noise.

This result is complemented by a negative result that asserts that in the stronger model of agnostic learning (wherein some arbitrary $\eta$-fraction of the data can have labels that disagree with $\mathrm{sign}(\langlew^*, x\rangle)$, without the stochastic structure imposed by Massart's condition). This result states that in such a setting, even learning a singleton out of a size-$n$ domain to excess error $1/4n$ requires $\Omega(n)$ TSQs, and thus shows that in general, one cannot use fewer than $\Omega(1/\varepsilon)$ TSQs to learn such a class (which further has a natural implication for halfspaces). The proof uses a reduction to distributed learning that is quite interesting.

**Strengths:**

Active learning of halfspaces under Massart noise is a challenging problem and of deep interest to the theoretical ML community, and thus this paper is certainly pertinent to the audience of Neurips. To my reading, the results are correct. Prior work is discussed in impressive detail, and the investigation of the paper is well contextualised.

Before offering more subjective comments, I do want to say that I am not an expert in this subfield of active learning, and my adjudication, especially of the novelty of the work, must be treated thus with care.

I find the results of this paper very interesting. I think that the TSQ setup does not appear to be _too_ powerful from the get go, and fits fairly well with some of the mistake-based query structures studied in prior work. The result captures the highly nontrivial setting of learning halfspaces without feature distribution assumptions, and the method proposed is elegant and clever. The writing of the paper is also excellent, and communicates subtle ideas in a clear manner, making the approach seem deceptively simple.

On the whole, I think this is a strong contribution to the literature.

**Weaknesses:**

I don't particularly see major weaknesses with the paper. However, I think that the main lacuna lies in the limited contextualisation of the a priori power of TSQs in the paper. While I appreciate the clear contextualiation of the investigation of query structures in active learning, I think the practicality of TSQs is not very clearly discussed in the paper. Mainly, the paper places discusses this in the context of mistake based queries, but clearly the TSQ setting is more onerous for a labeler, since they must compute $\phi$ over all examples in $S$ (rather than, e.g., simply declaring the first  mistake they find). I think that a frank discussion of how the authors think this query structure may be executed in practice would strengthen the contribution.

**Questions:**

-

**Limitations:**

I think this is fine viz the social impact stuff, but I suppose the contextualisation of TSQs I mentioned above can be viewed as a limitation that should be discussed a little more.

---

> ### Author Rebuttal · Authors · 2024-08-07
>
> We thank the reviewer for carefully reviewing our work and for the constructive feedbacks. Below is our response about the practicality of TSQs.
>
> > While I appreciate the clear contextualiation of the investigation of query structures in active learning, I think the practicality of TSQs is not very clearly discussed in the paper. Mainly, the paper places discusses this in the context of mistake based queries, but clearly the TSQ setting is more onerous for a labeler, since they must compute $\phi$ over all examples in $S$ (rather than, e.g., simply declaring the first mistake they find).
>
>
> Mistake-based query itself has already been useful in practice. For example, as mentioned in [BH12], mistake-based queries are used in Faces in Apple-iPhoto.
>
> From a theoretic perspective, TSQ is a natural generalization of previous query languages such as region queries, statistical queries, and label queries. In this paper, we did not focus on optimizing the structure of the queries we use in the algorithms. In fact, an efficient learning algorithm using TSQ may not exploit the full power of TSQ and thus depending on the application, one might be able to instead use other subclasses of TSQ that can be easily implemented. We believe that studying the tradeoff between the complexity of the query structure and the query complexity is very important and thus expect more future work done on designing simple and user-friendly TSQ for various learning problems.
>
> From an application perspective, we think learning with TSQ can also be used to formulate problems arising from practical applications as TSQ is a class of queries that can be computed in linear time.
> For example, there are many application such as sloving complicated tasks by interacting with LLMs, where a learning problem could be in general very hard to solve by an LLM, but human experts can break down the problems into sequential simple queries/questions such as TSQ that can be computed and verified fast by powerful models and thus can use LLM as a tool to solve complicated tasks. We will add more detailed discussions about these in the future version of the manuscript.
>
> >Reference
>
> >[BH12]Balcan, Maria Florina, and Steve Hanneke. "Robust interactive learning." Conference on Learning Theory. JMLR Workshop and Conference Proceedings, 2012.

---

> > ### Comment · Reviewer_VrwT · 2024-08-09
> >
> > Thanks for the clarifications, I hope that this interesting discussion does make it to the final version. I will keep my score.

---

### Author Rebuttal · Authors · 2024-08-07

We thank the reviewers for their time and effort in providing feedback. We are encouraged by the positive comments, and that all the reviewers appreciated the paper for the following (i) Novelty and interesting result (**VrwT,vqip,WAF8,DNzP**), (ii) technically clean and interesting (**VrwT, WAF8, DNzP**), (iii) clear presentation and well-motivated (**VrwT,vqip,WAF8,DNzP**) Below, we address the individual questions and comments by the reviewers separately.

---

### Decision · Program_Chairs · 2024-09-25

**Decision:**

Accept (spotlight)

**Comment:**

The reviewers unanimously agreed that the model considered in the paper is natural and that the results are interesting.